# Morphogeometric Evaluation of the Left Ventricle and Left Atrioventricular Ring in Dogs: A Computerized Anatomical Study

**DOI:** 10.3390/ani13121996

**Published:** 2023-06-15

**Authors:** Catarina Borges Cardoso, Cláudia Valéria Seullner Brandão, Paulo Sérgio Juliani, André Luis Filadelpho, Geovane José Pereira, Maria Lúcia Gomes Lourenço, Alessandre Hataka, Carlos Roberto Padovani

**Affiliations:** 1School of Veterinary Medicine and Animal Sciences–UNESP–Botucatu, São Paulo 18618-681, SP, Brazil; catarina.cardoso@unesp.br (C.B.C.); gj.pereira@unesp.br (G.J.P.); 2Department of Veterinary Surgery and Animal Reproduction, School of Veterinary Medicine and Animal–UNESP–Botucatu, São Paulo 18618-681, SP, Brazil; 3Cardiovascular Surgery Service, WeVets Veterinary Hospital, São Paulo 02511-000, SP, Brazil; 4Institute of Biosciences, Department of Anatomy–UNESP–Botucatu, São Paulo 18618-681, SP, Brazil; 5Department of Veterinary Clinics, School of Veterinary Medicine and Animal Sciences–UNESP–Botucatu, São Paulo 18618-681, SP, Brazil; maria-lucia.lourenco@unesp.br (M.L.G.L.);; 6Institute of Biosciences, Department of Biostatistics–UNESP–Botucatu, São Paulo 18618-681, SP, Brazil; cr.padovani@unesp.br

**Keywords:** ventricular geometry, anatomy, cardiac surgery, cardiac remodeling, myocardium

## Abstract

**Simple Summary:**

Veterinary cardiology recognizes the fundamental role of cardiac remodeling in the unfavorable progression of heart diseases, and this has been an object of study for therapies aimed at restoring the cardiac framework to its natural state. Therefore, this anatomical study proposes to obtain morphogeometric data of the left ventricle and left atrioventricular ring of normal dogs in order to describe reference values for the cardiac morphogeometric analysis in dogs.

**Abstract:**

In veterinary, there is scarce availability of morphogeometric studies in normal and remodeled hearts; furthermore, ventricular geometry acts as an indicator of cardiac function. It is a highly necessary field of knowledge for the development of therapeutic protocols, especially surgical ones. The objectives of this study were: to obtain measurements of the left atrioventricular valve ring and left ventricle, to analyze the proportionality between the segments of the left cardiac chamber of normal hearts and to describe reference values for morphogeometric analysis of the left ventricle. For this, 50 hearts from small (Group 1—G1) and medium to large (Group 2—G2) dogs were laminated in the apical, basal and equatorial segments, and submitted to computer analysis to identify the perimeter of each segment and the left atrioventricular ring, wall thickness and distance from the atrioventricular sulcus to the apex. The largest internal perimeter was that of the equatorial. The basal segment had the highest mean for ventral parietal wall thickness, suggesting greater contractile reserve at that location. Considering the proportionality relationships, there was no statistical difference between the intersegmental perimeter indices for the two groups. This suggests that despite the animals’ weight variations, the proportions between the left ventricular segments are maintained. Therefore, it is concluded that the data can be used as a standard of comparison for cardiac geometric assessments, as well as a basis for the development of therapeutic measures in the context of adverse cardiac remodeling.

## 1. Introduction

In human cardiology, numerous studies seeking to obtain a broad understanding of cardiac physiology were developed from the 1960s onwards, including the impact of ventricular geometry [1]. Although cardiac remodeling is a compensatory mechanism to pressure or volume overload, in the long term it is associated with heart failure, congestion and death, being an important component of cardiovascular disease [2,3]. The ventricular geometry is an indicator of left ventricle remodeling that may indirectly reflect impaired left ventricle function, and one of the best methods for studying the geometry of the ventricle is through the analysis of anatomical parts [1].

Left ventricular remodeling is the result of a combination of biochemical, physical, structural, molecular and geometric factors. Given that the left ventricular cavity loses its ellipsoid shape, tending to sphericity, functionality is impaired by increased parietal stress, increased oxygen consumption, subendocardial hypoperfusion and sustainment of neurohormonal and mechanical mechanisms [1,2,3,4]. 

Although advanced in medicine, the anatomical study of the geometry of the cardiac chambers is incipient in veterinary medicine; despite the contribution it can offer to the development and improvement of therapeutic protocols, such as surgical techniques, cardiac medical devices and pharmacological therapies [5,6]. 

Thus, this study aimed to obtain and characterize geometric data on the left ventricle of animals without cardiac alterations, in order to enable future morphogeometric assessment of the most varied heart disease in dogs. To the best of the authors’ awareness, such values have not yet been described in dogs, despite their great importance in medical cardiology, which stimulated the development of this study.

## 2. Materials and Methods

All experimental procedures were performed after approval by the Committee on Ethics in the Use of Animals at the School of Veterinary Medicine and Animal Sciences at the Paulista State University “Júlio de Mesquita Filho”, Botucatu Campus (Protocol CEUA 0053/2019).

The total of 50 hearts studied had normal macroscopic characteristics and originated from young adult or adult dogs, aged between one and five years and having been referred to the Small Animal Surgery Service at the Veterinary Hospital of the School of Veterinary Medicine and Animal Sciences (FMVZ), UNESP, the campus of Botucatu, or to the Municipal Kennel to the city government of Botucatu or Bertioga. The case selection period was one year (2019–2020). The hearts were collected according to the method described by Jones and Gleiser (1954) [7]. 

The hearts were obtained within a maximum period of 24 h post-mortem, together with conservation in a cold chamber (0 to 5 °C). After weighing and identification, the anatomical pieces were filled with cotton to maintain their original shape and fixed in a 10% formaldehyde solution, in accordance with other anatomical studies of the heart [6,8]. It is assumed that the retraction of tissues soaked in formaldehyde is uniform, maintaining the geometric framework [8]. 

The inclusion criteria adopted were hearts from dogs with no history of clinical signs related to heart disease, age between one and five years, and causa mortis not related to heart disease. Exclusion criteria included cardiac specimens presenting macroscopic alterations related to heart disease, those with inadequate macroscopic conservation, dogs for which cause of death could not be determined, or those from animals aged less than one year or more than five years. 

Two experimental groups were constituted, according to body weight: Group 1 (*n* = 20) comprised hearts from dogs weighing up to 10 kg and Group 2 (*n* = 30) hearts from dogs weighing more than 10 kg, without breed discrimination. 

### 2.1. Method of Preparation of Anatomical Specimens

Dissection and preparation were performed to obtain the digital images, which were subsequently submitted for analysis, according to the protocol used by Juliani et al. (2018) [6] in human hearts. 

Transversal cuts were made in each heart, using a knife on a flat surface, after fixing the specimen with the dorsal side facing upwards. To determine the cut sites, a master line was established on the dorsal cardiac face, parallel and close to the paraconal interventricular *sulcus* and perpendicular to the coronary *sulcus*, connecting this to the cardiac *apex* (Atrioventricular-apex distance), using a medical tape measure. From the master line, the proportionality criterion was set for cross sections of the left ventricle starting from the apex of the ventricle, at points relative to 20% (apical segment), 50% (equatorial segment) and 80% (basal segment), as shown in Figure 1. 

Atrioventricular valves were exposed and prepared for digital images by excision of the right and left atria in the basal cardiac region by delicate dissection (Figure 2).

### 2.2. Computerized Measurement 

After the sections were obtained, the segments were positioned to capture images by using a digital camera (Canon PowerShot ELPH 110 HS) fixed on a support positioned at a focal length of 25 cm. All images were submitted to computerized measurements of the chamber’s internal perimeter (Figure 3), atrioventricular ring perimeter and parietal thicknesses (Figure 4); using the software Image J, developed by the National Institutes of Health (USA). The images were measured three times by the same operator. The references for measuring ventral, dorsal, lateral and septal wall thicknesses were obtained using the protocol described (Juliani et al., 2018) [6] in medical cardiology, as shown in Figure 4, based on the limits of the septal wall. 

The following variables were obtained, in centimeters, according to the experimental groups: perimeter of the left atrioventricular ring, distance from the atrioventricular sulcus to the apex of the left ventricle (AV-AP Distance), the internal perimeter of the apical, equatorial and basal segments of the left ventricle and thickness of the ventral, lateral, dorsal and interventricular septum walls. Additionally, the bodily and heart weights were obtained, as well as the breed classification and sex of the dogs. 

### 2.3. Statistical Analysis

The comparison of the variables of internal perimeter and thicknesses evaluated in two independent groups and in three segments was performed using the technique of multivariate analysis of variance for the model of repeated measures in independent groups, complemented with the Bonferroni multiple comparisons test [9]. The Student’s t test for independent samples [10] was used to correlate the variables animal weight, heart weight, AV-AP distance and left atrioventricular ring perimeter, as well as the intersegmental perimeter index obtained for each group. The association between pairs of variables, in each group, was assessed by Pearson’s linear correlation coefficient, and the 95% confidence interval for the group mean was obtained according to Zar (2009) [10]. All discussions of the results were carried out considering a 5% significance level. Variables were assessed for normality using Shapiro-Wilk and Royston tests. 

#### 2.3.1. Comparison and Correlation of Measures

The perimeter of the left atrioventricular ring was correlated with the perimeters obtained for each segment. Segmental perimeters (apical, equatorial and basal) were correlated with each other. Segmental thicknesses of the ventral, dorsal, lateral and septal walls were compared within the same group. The measures were also compared between groups.

#### 2.3.2. Analysis of Left Ventricular Proportionality

In order to better understand the geometric behavior of a normal left ventricle, the perimeter data obtained were analyzed in each group using the intersegmental percentage relationship, expressed in 3 indexes: (1) Perimetral Percentage of the Base in relation to the Equator (PerB/PerE); (2) Perimetral Percentage of the Apex in relation to Ecuador (PerA/PerE); (3) Perimetral Percentage of Apex in relation to Base (PerA/PerB). 

## 3. Results

Group 1 consisted of 12 males (60%) and 8 females (40%), a total of 20 dogs, 75% of which were of mixed breed and 25% of the Pinscher and Shih Tzu breeds, with a mean age of 32.6 months and an average weight of 6.6 kg. Heart weights, ranging from 26 g to 87 g, presented a mean of 63.1 g. 

In Group 2, the hearts originated from 17 males (56.6%) and 13 females (43.3%), a total of 30 dogs, 83.33% of which were of mixed breed and 16.65% with only one representative (Labrador, Siberian Husky, Rottweiler, Pit Bull and Bull Terrier), with an average age of 48.1 months and an average weight of 19.1 kg. Heart weights ranged from 75 g to 362 g, with a mean of 176 g (Table 1). 

### 3.1. Variables Atrioventricular-Apex Distance (AV-AP) and Left Atrioventricular Ring Perimeter

Groups 1 and 2 presented respective mean AV-AP distances of 5.18 cm and 7.12 cm, as shown in Table 1 and Figure 5. The mean perimeter of the left atrioventricular ring was significantly greater in Group 2 as compared to Group 1 (*p* < 0.001). 

### 3.2. Segmental Internal Perimeters (Basal, Equatorial and Apical) of the Left Ventricle 

In Group 1, the equatorial segment had a greater internal perimeter, differing from the means found for the apical and basal segments (*p* < 0.001). In Group 2, the equatorial and basal segments did not differ, although both had greater perimeter than the internal apical segment (*p* < 0.001). 

The intergroup comparison showed a difference between the means observed in groups 1 and 2 (Apical: *p* < 0.05; Equatorial and Basal: *p* < 0.001). The data are displayed in Table 2. 

### 3.3. Thickness of the Ventral, Dorsal, Lateral, Septal Wall of the Left Ventricle

The ventral wall thickness was greater in the basal segment in both experimental groups (*p* < 0.001). The smallest measurements were found in the apical segment. Considering the dorsal wall thickness, a similar behavior was observed between the equatorial and basal segments, both being greater than the dorsal wall thickness in the apical segment (*p* < 0.001). Yet the lateral wall thickness, in Group 2, was greater in the equatorial segment, while such measurements in the apical and basal segments were equivalent (*p* < 0.001). However, in Group 1, there was no significant difference between the heart segments (*p* > 0.05). Data for wall thicknesses are shown in Table 3, Table 4, Table 5 and Table 6 and Figure 6. 

Measurements of the variable septal thickness indicate that, for Group 1, the mean of the apical segment is smaller (*p* < 0.001), with no difference between the basal and equatorial segments. In Group 2, mean thickness was similar between the apical and basal segments, and greater (Table 6) for the equatorial segment (*p* < 0.001). 

Thus, the intragroup behavior of the means was similar in relation to the thickness of the lateral and septal walls in Group 2. 

The intergroup comparison evidenced a statistical difference between the thicknesses of all the walls of the left ventricle in all segments, except for the thickness of the septal wall in the basal segment (*p* > 0.05). 

### 3.4. Confidence Interval 95%

The 95% confidence interval was described in order to establish a reference value for the variables, and were expressed in Table 7.

### 3.5. Analysis of the Correlation between the Segmental Perimeters of the Left Ventricle 

The linear association measures between the pairs of variables revealed a positive and moderate correlation (r = 0.463) between the apical and basal internal perimeter in Group 1 (*p* < 0.05). In Group 2, a significant positive association was observed between the variables left atrioventricular ring perimeter and basal internal perimeter (r = 0.526; *p* < 0.01) and equatorial internal perimeter in relation to the basal internal perimeter (r = 0.396; *p* < 0.05). All correlation indices between the segmental perimeters of the left ventricle are described in Table 8.

### 3.6. Analysis of the Intersegmental Perimeter Index

All perimeter indices evaluated (PerA/PerB; PerB/PerE; PerA/PerE) were constant between the two groups (Table 9), that is, there was no statistical difference between the percentage relationships between the ventricular segments (*p* > 0.05) in animals of different body weights.

## 4. Discussion

The study characterized the geometric morphometry of the left ventricle of dogs, in normal anatomical specimens, with special interest in the apical, equatorial and basal segments and their relationships; considering their qualification in two dog weight ranges, divided into dogs weighing up to 10 kg and those above 10 kg. Such data could indicate ventricle function and prognosis in alive animals, being important in cardiology, but are still scarce in Veterinary Medicine. 

The study of anatomical specimens is a reliable source of morphogeometric data [1]. The knowledge of reference values is essential for evaluating alterations in the shape of the heart in the most varied of heart diseases in dogs. That said, the geometry applied to the cardiac configuration allows the assigning of numerical values to the differences observed between specimens in different clinical conditions and, therefore, the inferring of their causes and consequences. 

In dogs, studies have been conducted on cardiac morphometry [11,12] and on the use of magnetic resonance imaging [13] and echocardiographic indices [14,15] to assess cardiac morphology. However, there is an insufficiency of studies that assess the normal cardiac geometry of dogs considering the basal, equatorial and apical ventricular segments and their relationships with each other and with the left atrioventricular ring. 

The criterion for dividing the groups by weight was based on the prevalence of small animals in myxomatous valve degeneration [16] and medium and large animals in dilated cardiomyopathy [17], respectively, aiming to generate data valid for comparative analyses with the pathological groups in future studies. 

Heart weight is a variable associated with muscle hypertrophy and may act as a diagnostic factor for heart disease in necropsy examinations [18]. The mean cardiac weight observed corresponded to 0.95% of the body weight in dogs in G1 and 0.91% in G2, a range similar to 0.6 to 1.1% of body weight described for dogs without macroscopic features of heart disease [11,12], thus reinforcing the method for including normal hearts in this study. 

In the present study, the animal’s sex was not taken into account for the analysis of the variables, an option based on the literature that evidences an absence of differences in cardiac morphometry indices in male and female dogs [11,12]. 

During the analysis of the results, the studied variables were observed separately and as a whole, associating them in order to understand their relationships as an organ. About basal, equatorial and apical perimeters, a study indicated for normal human hearts that the apical perimeter mean is statistically lower compared to the other segments, while the basal and equatorial segments perimeters present coincident means [6]. The analysis of segmental perimeters for healthy dogs from a cardiologic point of view shows similar behavior in Group 2 (>10 kg), in which the apical segment has a smaller perimeter (*p* < 0.001) and the basal and equatorial segments are statistically equivalent. The statistical difference found between the perimeter of the basal and equatorial segments in Group 1 may be related to the variability between cardiac silhouettes and thoracic conformations, similar to physiologic differences in normal Vertebral Heart Size (VHS) values observed in dogs of different breeds [19]. 

The left ventricular circumference in dog hearts in systole and diastole was studied via echocardiography, which obtained means close to those observed in the present study when comparing perimeters in systole [20]. Circumferences in diastole, on the other hand, are larger, which may suggest the influence of the cardiac cycle at the time of the animal’s death in studies using anatomical specimens. 

The measures of association were calculated with the aim of verifying whether the segments maintain mutually dependent relationships. In Group 2, the association between equatorial and basal perimeters was identified, as previously described in normal human hearts [6]. 

The slight but correlated difference between equatorial and basal perimeters in normal hearts suggests an insignificant angulation in this left ventricular outflow tract. According to previous studies, [6,21] ventricular systole can promote a vector resulting from longitudinal (apico-basal contraction) and transversal (apico-equatorial and equatorial-basal circumferential contraction) forces, with the circumferential contraction of the equatorial-basal region being responsible for the generation of a resulting vector perpendicular to the others towards the exit route. Alterations in the equatorial-basal arrangement of the ventricles due to remodeling can directly impact the efficiency of the heart as a pump, generating turbulent flow in the left ventricular outflow tract [6,22]. 

The significant moderate correlation observed between the apical and basal internal perimeters in Group 1 was also found in normal human hearts, but at a lower intensity [6]. The evaluation of the perimeter of the left atrioventricular ring identified values for the two groups, which differed statistically, as expected because they were groups composed of animals of different sizes. Furthermore, there was a moderate but significant positive correlation (*p* < 0.01) between the left atrioventricular ring perimeter and internal basal perimeter in Group 2. Future studies will be able to prove whether this correlation is maintained in remodeled hearts. It was observed in cardiac remodeling of human hearts that there is dilation of the left atrioventricular ring; however, this does not have a dependent relationship with the dilation of ventricular segments [6,23]. In addition, several surgical procedures aimed at restoring the mitral annulus to a normal state are described in the medical literature, given the great importance of this structure in the course of cardiac remodeling [24]. 

Analyzing the parietal thicknesses, it is noted that the ventral wall has a greater thickness in the basal segment, while the dorsal walls of this same segment present a mean equal to that of the equatorial segment. The septal and lateral walls of the basal segment, in turn, maintain similarity to the equatorial segment in Group 1 and to the apical segment in Group 2. 

The thickness pattern for the ventral wall in the basal segment is similar to that observed in normal and remodeled human hearts [6]; however, in these, the thickness of the dorsal wall, denominated posterior in humans, is also significantly greater in the basal segment, which was not observed in the hearts of either group in the present study. 

The distribution of parietal stress can be deduced from myocardial thickness patterns [4]. One hypothesis to be considered based on the results obtained is that the ventral wall in its portion of the basal segment is exposed to greater stresses due to the effect of gravity, since dogs are quadrupedal animals. 

Also, in relation to parietal thicknesses, the dorsal wall thickness in the equatorial segment had an average of 1.2 cm in Group 1 and 1.59 cm in Group 2, with data from Group 1 corroborating the data observed in echocardiographic studies of beagles weighing an average of 8.92 kg without cardiac alterations in a study [20].

In the study in question, the distance between the atrioventricular sulcus and the apex of the left ventricle was characterized in both groups. These data are important to assess the occurrence of longitudinal dilation in remodeled hearts in future research studies, a phenomenon associated with the redirecting of muscle fibers into a more longitudinal configuration to maintain the optimized cardiac function acquired by the helical configuration of muscle fibers [25,26]. 

Within each group evaluated, a wide weight range was observed among the individuals. However, from the analysis of the intersegmental perimeter index expressed as a percentage, it is possible to see that, although the weights of the individuals and, therefore, the dimensions of the left ventricle in absolute values vary, the proportional relations between the segments remain between the two groups. As far as the authors are aware, such indices of proportion between the segments of the left ventricle had not been established in dogs. These data can support the strategic planning of techniques to contain cardiac remodeling, allow future assessments of the proportionality of left ventricular dilation and generate a structural framework for the basis of studies on the bioengineering of cardiac tissues, through three-dimensional printing of organs [27]. 

In the region where the study was conducted, the prevalence of mixed breed dogs was high, making it impossible to standardize the body and, especially, the thoracic conformation of these dogs, despite their segregation into groups by weight ranges. This fact constitutes a limitation of the study, which allows us to suggest that future research studies focus on aspects of greater segmentation between sizes and breeds of dogs. However, this context expresses a reality in the analyzed regional population. 

It is believed that the measurement of morphogeometric data from normal dogs’ hearts contributes comparative parameters for specimens affected by heart disease and, therefore, subject to remodeling. There is a vast literature linking the shape alteration arising from remodeling adverse to disease progression and heart failure functional class [2,3,28]. Due to the significant importance of cardiac architecture, several therapeutic strategies seek to restore it, aiming at better ventricular function [5,25,26].

## 5. Conclusions

Given the above findings, it is concluded that, for the population studied, the perimeter proportion between the segments of the left ventricle is maintained despite the variation in weight between individuals, and that the data obtained on parietal thicknesses, perimeter of ventricular segments, atrioventricular-apex distance and perimeter of the left atrioventricular ring characterize possible reference values for future research on the distribution and clinical implications for cardiac remodeling in dogs. 

## Figures and Tables

**Figure 1 animals-13-01996-f001:**
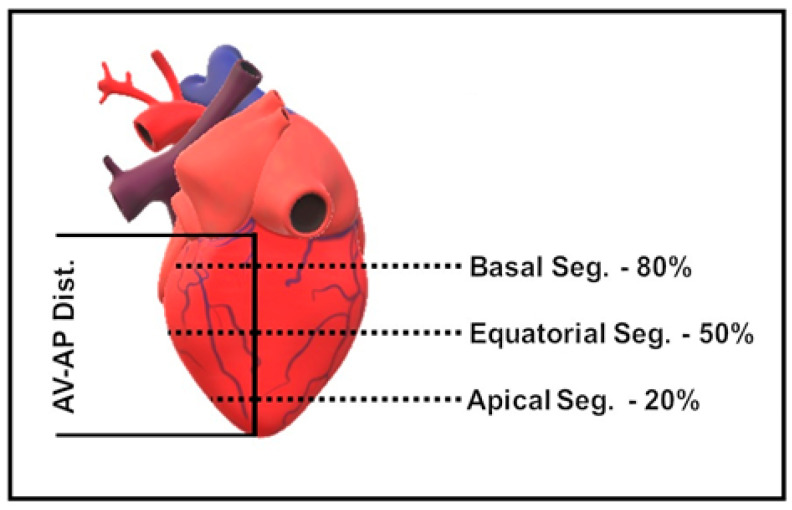
Schematic representation of the lamination points, dashed line, of the left ventricle. AV-AP Dist.: Atrioventricular-apex distance; Basal Seg.: Basal Segment; Equatorial Seg.: Equatorial Segment; Apical Seg.: Apical Segment.

**Figure 2 animals-13-01996-f002:**
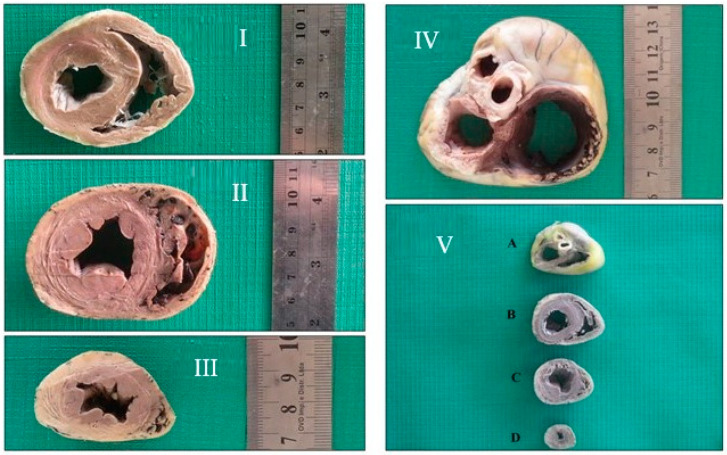
Photographic image of the cranial face of the basal (**I**), equatorial (**II**) and apical (**III**) segments of a dog’s left ventricle (Group 2). (**IV**)—Cranial face of a dog’s left atrioventricular region (Group 2). (**V**)—Photographic image of a dog’s left ventricle segments (Group 1), in different regions. A. Cardiac base region, B. Basal segment, C. Equatorial segment, D. Apical segment.

**Figure 3 animals-13-01996-f003:**
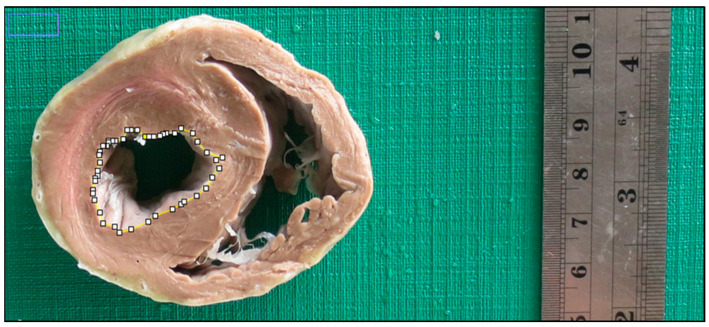
Photographic image of internal perimeter measurement of basal segment of the left ventricle of a dog (Group 2), utilizing the software Image J.

**Figure 4 animals-13-01996-f004:**
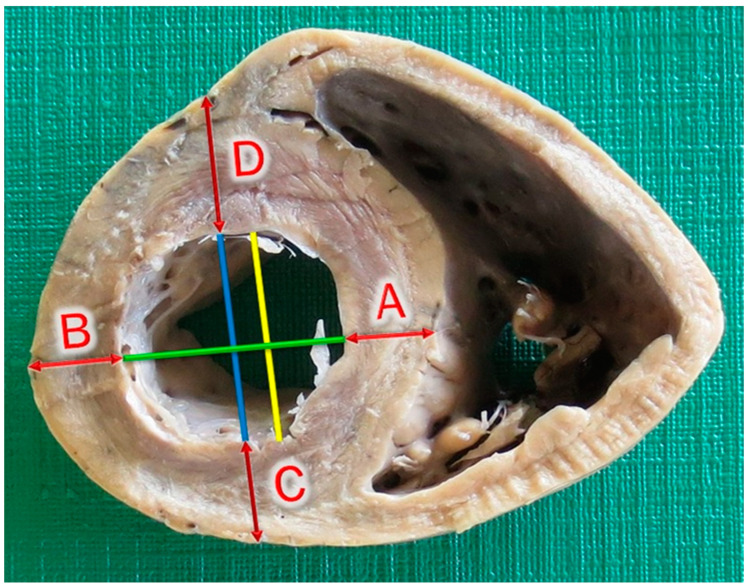
Demonstration of the method for determining measurement points of parietal thicknesses. Yellow line-Limits of septal wall. Green line–Obtained from the middle distance of the yellow line; guideline for points of septal (A) and lateral (B) walls. Blue line–Obtained from the middle distance of the green line; guideline for points of dorsal (C) and ventral (D) wall.

**Figure 5 animals-13-01996-f005:**
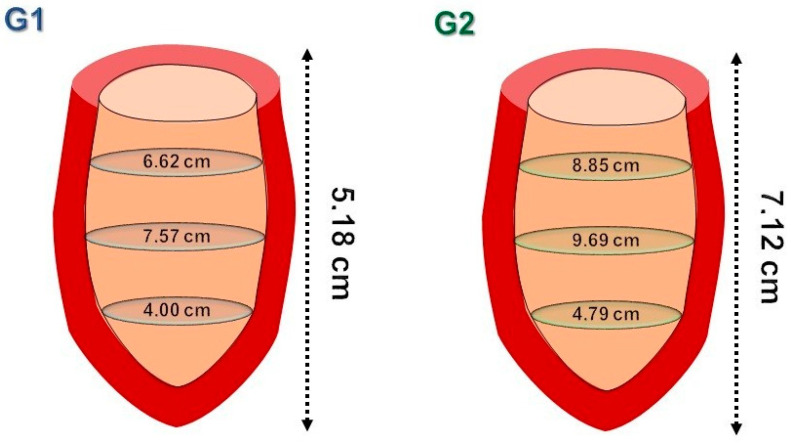
Schematic representation of the mean values of segmental internal perimeters and AV-AP distance in the study population, considering Group 1 (**G1**) and Group 2 (**G2**). **G1**—dogs weighing up to 10 kg; **G2**—dogs above 10 kg.

**Figure 6 animals-13-01996-f006:**
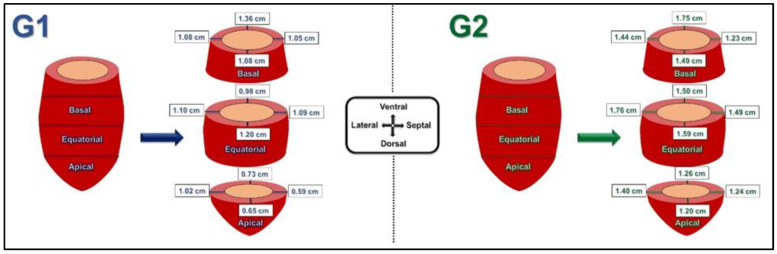
Schematic representation of the mean values of the thicknesses of the septal, lateral, dorsal and ventral walls in the basal, equatorial and apical segments in the study population, considering Group 1 (**G1**) and Group 2 (**G2**). **G1**—dogs weighing up to 10 kg; **G2**—dogs over 10 kg.

**Table 1 animals-13-01996-t001:** Mean and standard deviation of the variables weight, heart weight, atrioventricular-apex distance (AV-AP Dist.), left atrioventricular ring perimeter, for Group 1 (*n* = 20) and Group 2 (*n* = 30).

	Group		
Variable	G1	G2	*p*-Value
Body weight (kg)	6.6 (2.45)	19.1 (9.41)	*p* < 0.001
Heart weight (g)	63.1 (19.78)	176 (63.72)	*p* < 0.001
Dist. AV-AP	5.18 (0.55)	7.12 (0.90)	*p* < 0.001
Left Atrioventricular Perimeter	4.75 (0.91)	6.33 (1.17)	*p* < 0.001

**Table 2 animals-13-01996-t002:** Mean values (cm) for the internal perimeter of the apical, equatorial and basal heart segments in dogs of Group 1 (*n* = 20) and Group 2 (*n* = 30).

Group	Segments			
	Apical	Equatorial	Basal	*p*-Value
G1	4.00 (1.19) ^a,^*	7.57 (1.58) ^c^	6.62 (0.96) ^b^	*p* < 0.001
G2	4.79 (1.45) ^a^	9.69 (1.61) ^b^	8.85 (1.70) ^b^	*p* < 0.001
*p*-value	*p* < 0.05	*p* < 0.001	*p* < 0.001	

* For the intragroup analysis (lines): two means followed by the same lower-case letter do not differ (*p* > 0.05) from each other.

**Table 3 animals-13-01996-t003:** Mean values (cm) for the thickness of the ventral wall in the apical, equatorial and basal segments of the left ventricle of dogs, for groups 1 (*n* = 20) and Group 2 (*n* = 30).

Group	Segments			
	Apical	Equatorial	Basal	*p*-Value
G1	0.73 (0.27) ^a,^*	0.98 (0.32) ^b^	1.36 (0.38) ^c^	*p* < 0.001
G2	1.26 (0.32) ^a^	1.50 (0.36) ^b^	1.75 (0.55) ^c^	*p* < 0.001
*p*-value	*p* < 0.001	*p* < 0.001	*p* < 0.01	

* For the intragroup analysis (lines): two means followed by the same lower-case letter do not differ (*p* > 0.05) from each other.

**Table 4 animals-13-01996-t004:** Mean values (cm) for the thickness of the dorsal wall in apical, equatorial and basal segments of the left ventricle from dogs of Group 1 (*n* = 20) and Group 2 (*n* = 30).

Group	Segments			
	Apical	Equatorial	Basal	*p*-Value
G1	0.65 (0.30) ^a,^*	1.20 (0.48) ^b^	1.08 (0.32) ^b^	*p* < 0.001
G2	1.20 (0.27) ^a^	1.59 (0.43) ^b^	1.49 (0.29) ^b^	*p* < 0.001
*p*-value	*p* < 0.001	*p* < 0.01	*p* < 0.001	

* For the intragroup analysis (lines): two means followed by the same lower-case letter do not differ (*p* > 0.05) from each other.

**Table 5 animals-13-01996-t005:** Mean values (cm) for the thickness of the lateral wall in apical, equatorial and basal segments of the left ventricle from dogs of Group 1 (*n* = 20) and Group 2 (*n* = 30).

Group	Segments			
	Apical	Equatorial	Basal	*p*-Value
G1	1.02 (0.44)	1.10 (0.44)	1.08 (0.39)	*p* > 0.05
G2	1.40 (0.43) ^a,^*	1.76 (0.40) ^b^	1.44 (0.34) ^a^	*p* < 0.001
*p*-value	*p* < 0.01	*p* < 0.001	*p* < 0.005	

* For the intragroup analysis (lines): two means followed by the same lower-case letter do not differ (*p* > 0.05) from each other.

**Table 6 animals-13-01996-t006:** Mean values (cm) for the thickness of the septal wall in apical, equatorial and basal left ventricle segments from dogs of Group 1 (*n* = 20) and Group 2 (*n* = 30).

Group	Segments			
	Apical	Equatorial	Basal	*p*-Value
G1	0.59 (0.27) ^a,^*	1.09 (0.35) ^b^	1.05 (0.26) ^b^	*p* < 0.001
G2	1.24 (0.35) ^a^	1.49 (0.28) ^b^	1.23 (0.34) ^a^	*p* < 0.001
*p*-value	*p* < 0.001	*p* < 0.001	*p* > 0.05	

* For the intragroup analysis (lines): two means followed by the same lower-case letter do not differ (*p* > 0.05) from each other.

**Table 7 animals-13-01996-t007:** 95% confidence limits of the different variables evaluated in the left ventricle of dogs, in groups 1 (*n* = 20) and 2 (*n* = 30).

Variable	Group 1		Group 2	
	LL	UP	LL	UP
AV-AP Distance	4.94	5.42	6.81	7.12
Left Atrioventricular Ring Perimeter	4.36	5.14	5.92	6.33
Apical Internal Perimeter	3.47	4.53	4.28	4.79
Equatorial Internal Perimeter	6.88	8.26	9.12	9.69
Basal Internal Perimeter	6.21	7.03	8.24	8.85
Ventral Apical Thickness	0.61	0.85	1.14	1.26
Ventral Equatorial Thickness	0.84	1.12	1.36	1.50
Ventral Basal Thickness	1.20	1.52	1.55	1.75
Dorsal Apical Thickness	0.51	0.79	1.10	1.20
Dorsal Equatorial Thickness	0.98	1.42	1.43	1.59
Dorsal Basal Thickness	0.94	1.22	1.39	1.49
Lateral Apical Thickness	0.82	1.22	1.24	1.40
Lateral Equatorial Thickness	0.90	1.30	1.62	1.76
Lateral Basal Thickness	0.90	1.26	1.32	1.44
Septal Apical Thickness	0.47	0.71	1.12	1.24
Septal Equatorial Thickness	0.93	1.25	1.39	1.49
Septal Basal Thickness	0.93	1.17	1.11	1.23

AV-AV Dist.: Atrioventricular-Apex Distance. LL Lower limit. UP: Upper limit.

**Table 8 animals-13-01996-t008:** Measurement of linear association between left ventricle internal perimeters and left atrioventricular ring perimeter of dogs from Group 1 (*n* = 20) and Group 2 (*n* = 30).

	Group	
Association	G1	G2
Left Atrioventricular ring Perimeter vs. Apical Perimeter	0.135 (*p* > 0.05)	0.113 (*p* > 0.05)
Left Atrioventricular ring Perimeter vs. Equatorial Perimeter	0.326 (*p* > 0.05)	0.142 (*p* > 0.05)
Left Atrioventricular ring Perimeter vs. Basal Perimeter	0.362 (*p* > 0.05)	0.526 (*p* < 0.01)
Apical Perimeter vs. Equatorial Perimeter	0.370 (*p* > 0.05)	0.054 (*p* > 0.05)
Apical Perimeter vs. Basal Perimeter	0.463 (*p* < 0.05)	0.033 (*p* > 0.05)
Equatorial Perimeter vs. Basal Perimeter	0.423 (*p* > 0.05)	0.396 (*p* < 0.05)

**Table 9 animals-13-01996-t009:** Mean and standard deviation of Perimeter Segmental Indices in the left ventricle of dogs from Group 1 (*n* = 20) and Group 2 (*n* = 30).

Perimeter Indices	Group		
	G1	G2	*p*-Value
PerA/PerB	60.41 (15.23)	55.59 (16.60)	*p* > 0.05
PerB/PerE	92.21 (23.04)	92.48 (17.02)	*p* > 0.05
PerA/PerE	54.09 (15.85)	50.67 (15.89)	*p* > 0.05

## Data Availability

Data available only upon request due to privacy/ethical restrictions.

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
