# Peer review of "Morphogeometric Evaluation of the Left Ventricle and Left Atrioventricular Ring in Dogs: A Computerized Anatomical Study"

_animals, 2023, doi:10.3390/ani13121996_

Round 1

Author Response

Dear revisers,                                                           

We really appreciate the contributions and guidance regarding our study; there are important considerations, and certainly, they will improve our manuscript greatly.

The version of the manuscript attached contains the corrections we have made using the “Track Changes” function as required. In this letter, we are going to detail some specific topics.

Reviser 1:

  • We thank and accept the suggestion about the title regarding the removal of the word “healthy” and the use of precise terms to describe that is an anatomical/post mortem Therefore, the new title is Morphogeometric evaluation of the left ventricle and left atrioventricular ring in dogs: a computerized anatomical study. We have also changed this in the simple summary, abstract, and the whole text.
  • About the sample normality, we have included the tests used to assess the data normality in paragraph 2.3 (Statistical analysis). On this content, here is the explanation provided by our statistician:
    • For the test using a univariate procedure, the Shapiro-Wilk test was applied to verify adherence to normality. For the statistical test using a multivariate procedure, the Royston test was applied to verify the multinormal adherence of the response vectors of the data.
  • The terms dorsal/ventral and lateral/septal related to dog hearts were used in accordance with the Nomina Anatomica Veterinaria, 6th edition, 2017, which brings the information that the terms anterior/posterior cannot be applied to quadrupeds.
  • Line 48/49: We thank and accept the corrections.
  • Line 57-58: We agree with that and we would like to acknowledge the contribution of this comment.
  • Line 66/72/86: We thank and accept the corrections.
  • Line 98: We have included the methodology used to measure the AV-AP distance in the Material and Methods section.
  • Figure 1/Line 107/145/150: We thank and accept the corrections.
  • Figure 2: The term cranial was used based on Nomina Anatomica Veterinaria, 6th edition, 2017, according to which it refers to direction. We have corrected the legend of the Figure 3, which is indeed a basal segment, as it was well noted by the reviser.
  • Figure 4: The yellow line represents the limits of the septal wall, and it is a reference to determine the green line in its middle. The middle distance of the green line is a reference to the blue line. Juliani et al. (2018) described this methodology in the assessment of wall thickness in normal human hearts. We appreciate and accept the correction, and have improved our explanation in the Figure 4 legend.
  • The statistical analysis paragraph was reorganized as suggested.
  • We agree that Paragraph 2.4 was redundant, so it was removed.
  • Paragraph 5: More specific data about the annulus measurement was included in the Material and Methods
  • Paragraph 2.6/Line 171/172-173/Paragraph 3.1/Line 181/182: We thank and accept the corrections.
  • Table 1/3/4/5/7/8/9: We thank and accept the corrections.
  • Figure 6: We thank and accept the corrections.
  • Line 269/295/299/306/313/325/331: We thank and accept the corrections.
  • The same operator made all the measurements three time. We have included this information in the text.
  • The period of case selection was 1 year (2019-2020), and this information was also included in the text.
  • Paragraph 3.2: We have rewritten this paragraph to make it clearer for the reader.
  • Figure 5: We thank and accept the corrections.
  • Line 208: We have rewritten this paragraph to make it clearer for the reader.
  • Table 2-6: The lower-case letters refers to the intragroup analysis with the aim to determine which segment presents statistical difference. The intergroup analysis is not defined by the lower-case letter, instead of that, the p-value at the last line of the table represents the presence or not of statistical difference between groups 1 and 2. We have supposed that it became unclear and confusing, so, we have added an explanation in the legend of the table.

Reviewer 2 Report

The paper presents interesting results of morphometric studies of the left ventricle and the atrioventricular orifice of the heart of dogs. However, it is difficult to understand why the authors did not do this for the other chambers. The more that the volume of the manuscript is quite modest, the number of words (2500) does not meet the minimum requirements for an article in the journal Animals. The term "computerized study" in the title is misleading and redundant. Anatomical terminology in many places is quite loose and deviates from the standard contained, for example, in Nomina Anatomica Veterinaria. For example, authors use "atrioventricular sulcus" instead of "coronary sulcus", "apex of the ventricle" instead of "cardiac apex" or "apex cordis". In lines 96-97 "interventricular sulcus" probably refers to "paraconal atrioventricular sulcus". My minor objection is not to use Italic in Latin names, I don't understand what "pieces" means in lines 76 and 78. "digital images by using digital camera" (lines 120-121) seems redundant.

Author Response

Dear revisers,                                                           

We really appreciate the contributions and guidance regarding our study; there are important considerations, and certainly, they will improve our manuscript greatly.

The version of the manuscript attached contains the corrections we have made using the “Track Changes” function as required. In this letter, we are going to detail some specific topics.

Reviser 2:

  • The failure of cardiac pump is highly associated with insufficiency of the left ventricle because of its important role on the cardiac remodeling. Dogs are commonly affected by myxomatous degeneration of the mitral valve; therefore, we have decided to start our study by the left ventricle and the left atrioventricular ring. We acknowledge the importance of this kind of study of the other cardiac chambers, and they will probably be part of our future studies.
  • Anatomical terminology: We thank and accept the corrections.
  • Line 76-78: The term “pieces” refers to the anatomical parts, like the heart. We have included the word anatomical to the sentence to make it clearer for the reader.
  • Line 120-121: We thank and accept the correction.
  • Number of words: The main text has 4,440 word in accordance with instructions for the authors (minimum around 4,000 words) for Original Research. We have supposed that the number of words cited by the reviser refers to Case Report, but we are available for more information about this topic.

We are very grateful for the correction and revisers´ contribution to our text and to the knowledge of our research group. We foresee that the publication of this manuscript is of high interest to the readership of the journal, and we have received and accepted all the suggestions with great enthusiasm.

Round 2

Reviewer 2 Report

The manuscript has been improved by the authors. I recommend it for publication in Animals.